# Propagating Knowledge Updates to LMs Through Distillation

**Shankar Padmanabhan, Yasumasa Onoe, Michael J.Q. Zhang, Greg Durrett, Eunsol Choi**
Department of Computer Science
The University of Texas at Austin
shankarpadmanabhan@utexas.edu

## Abstract

Modern language models have the capacity to store and use immense amounts of knowledge about real-world entities, but it remains unclear how to update such knowledge stored in model parameters. While prior methods for updating knowledge in LMs successfully inject atomic facts, updated LMs fail to make inferences based on injected facts. In this work, we demonstrate that a context distillation-based approach can both impart knowledge about entities *and* propagate that knowledge to enable broader inferences. Our approach consists of two stages: transfer set generation and distillation on the transfer set. We first generate a transfer set by prompting a language model to generate continuations from the entity definition. Then, we update the model parameters so that the distribution of the LM (the 'student') matches the distribution of the LM conditioned on the definition (the 'teacher') on the transfer set. Our experiments demonstrate that this approach is more effective at propagating knowledge updates than fine-tuning and other gradient-based knowledge-editing methods. Moreover, it does not compromise performance in other contexts, even when injecting the definitions of up to 150 entities at once.

## 1   Introduction

As large language models (LLMs) are used for a wider variety of applications, it is crucial to ensure that they contain up-to-date information about the world. One potential solution is retrieval augmentation, which prepends retrieved texts to the language model's context [20, 29, 35, 34]. However, this raises inference costs and becomes impractical when updating large amounts of information. An alternative approach, and our goal in this work, is to internalize the new knowledge into the language model via parameter updates [36, 44, 8, 26, 22, 12].

Recent work on injecting LLMs with information about emerging entities [32] demonstrates that updating parameters effectively enables models to acquire updated facts (*Rishi Sunak is the prime minister of the UK*), but struggles to teach models how to *propagate* this knowledge, or make inferences based on it *(what might Rishi Sunak do tomorrow?)*. This contrasts with results from retrieval augmentation [20, 35] and chain-of-thought prompting [40], which show that LLMs can make such inferences when information is placed in the prompt.

This work aims to bridge the gap between the two approaches in knowledge injection. We use a form of knowledge distillation [13] called context distillation [1] that updates an LM to act like it is conditioned on a given context, even when that context is not shown. Our approach consists of two steps: transfer set generation and distillation on the generated transfer set. The transfer set consists of continuations of the entity definition sentence generated by prompting a language model. To distill on this transfer set, we minimize the Kullback–Leibler (KL) divergence between the model's predictions

37th Conference on Neural Information Processing Systems (NeurIPS 2023).

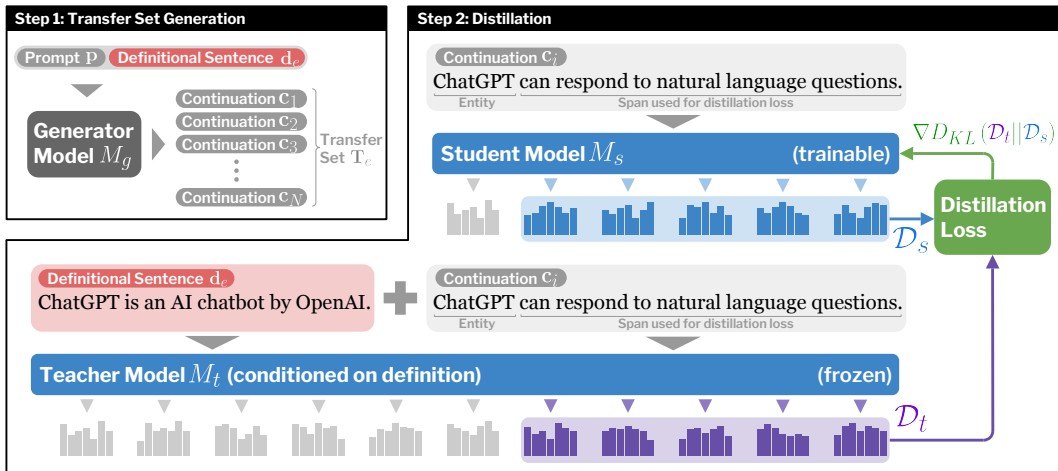

Figure 1: Overview of our distillation approach. Our goal is to inject the entity definition ($\mathbf{d}_e$) into the student model ($M_s$) and propagate it to make inferences based on the injected knowledge. This example uses *ChatGPT* as a new entity. We first generate a set of continuations of the entity's definition using a generator model (Step 1), then use these to distill the information from definition into the student model via a KL loss between the conditioned and unconditioned models (Step 2); see Section 3 for formulation.

on the transfer set when it conditions on the definition (the "teacher" for distillation) and when it does not (the "student", or the language model itself). Figure 1 shows this approach.

We evaluate our approach on two knowledge propagation benchmarks: ENTITY INFERENCES [32] and Entity Cloze by Date (ECBD) [31]. We evaluate on three language models and find that our distillation approach outperforms fine-tuning and prior editing methods (MEND [26] and MEMIT [23]) across all models. To investigate the robustness of our approach, we present an ablation study focusing on the design choices during transfer set construction. Encouragingly, we find that distilling on transfer sets constructed from the base language model itself is competitive with those generated by a much larger model (GPT-3.5). This demonstrates that context distillation does not rely on distilling from a larger model, and that our approach can work across a range of model sizes. Finally, we show that our approach can be scaled to inject larger amounts of information at once: we can inject over 100 new entities into a language model with minimal performance degradation, suggesting that the distillation process performs relatively targeted editing even without additional objectives to ensure specificity as in past methods [22, 23].

To summarize, we present a new approach for propagating injected knowledge. We show that a knowledge distillation technique can effectively impart and propagate knowledge from entity definitions into the parameters of a pre-trained language model, compared to existing knowledge editing methods. Yet, we observe robust gap between providing information in-context and parameter updating methods, leaving ample room for future work. Our code and data are available at `https://github.com/shankarp8/knowledge_distillation`.

## 2 Background and Task Setup

### 2.1 Motivating Example

Figure 1 shows a motivating example. An LM trained on text collected prior to November 2022 will not have specific knowledge about what ChatGPT is, as ChatGPT was introduced after that time. Past retrieval-augmented generation methods [20] have shown that conditioning on information about this entity can lead to lower perplexities when evaluating on sentences like *ChatGPT can respond to natural language questions* [35, 32]. For example, the model assigns a higher likelihood to tokens like *respond* given the knowledge that ChatGPT is a chatbot.

Our approach relies on teaching a "student model" (the LM itself) to match the next-token distributions given by the model conditioned on the definition sentence *even when the definition sentence is not shown*. We do this via a distillation process on a set of *continuations*, or sampled sentences following

the definition. We impose a KL penalty between the student and teacher distributions of a set of target tokens, namely all those occurring after *ChatGPT* in the continuation. Because the distillation process does not make updates on tokens where the teacher and student have the same distribution (zero KL), only tokens that are in some way predictable from the definition drive parameter updates (see Section 7 for discussion).

## 2.2 Task Setup

We refer to language models $M$ as $M(\mathbf{x}) \to \mathcal{D}(\mathcal{V})$, mapping an input context $\mathbf{x} = (x_1, \ldots, x_n)$ to a next-word distribution $\mathcal{D}(\mathcal{V}) = p(\cdot \mid x_1, \ldots, x_n)$ over a vocabulary $\mathcal{V}$. We will also use $M(\mathbf{x}) \to \mathcal{D}(\mathcal{V})_{1,\ldots,n}$ to represent the collection of distributions after each prefix of $\mathbf{x}$, which is a standard operation used in language model training. To update knowledge in the base language model $M_{\text{Base}}$, definitional information $\mathbf{d}_e = (d_1, \ldots, d_m)$ for an entity $e$ is provided. We will use $e$ both as an indicator and also a reference to the entity name string (e.g., *ChatGPT* in Figure 1). Our goal is to update $M_{\text{Base}}$ to $M_s$ so that it "knows" $\mathbf{d}_e$, by matching $M_s(\mathbf{x})$ with $M_t(\mathbf{x} \mid \mathbf{d}_e)$ (the teacher model) as closely as possible with our distillation scheme, when $\mathbf{x}$ is relevant to entity $e$. We set the teacher model $M_t$ to be a copy of $M_s$.

We evaluate on two factors. First, **propagation success** measures how well the updated language model $M_s$ acquired information about $\mathbf{d}_e$ to make correct inferences in probe sentences. Crucially, our evaluation here is not just a narrow notion of whether a specific fact is injected [44, 8, 26, 22, inter alia], but captures the model's ability to make inferences on it [31, 32]. Second, **specificity** evaluates whether the predictions of the LM on other contexts are altered as in prior work [8, 26, 22, 23]. Ideally, edits should not impact inferences on examples unrelated to the edit.

## 2.3 Related work

**Knowledge distillation** We are not aware of prior work that uses distillation for knowledge editing. Our use of context distillation is most similar to Askell et al.'s alignment work [1]; however, they use it in a phase roughly analogous to RLHF and use a generic transfer set sampled from the language model training corpus. Our work is also related to prompt injection [5], which examines tasks like distilling a persona-conditioned language model. Unlike their work, we do not train a task-specific model to generate examples for distillation. Instead, we simply prompt existing LMs. Furthermore, while they aim to have a model memorize a particular prompt, we focus on general knowledge updates and inferences based on those. Other work has used distillation techniques for "gisting" to make shorter prompts [28] or to distill reasoning processes [37]. Similar approaches as our continuation sampling have been used for example extrapolation [19] to generate training datasets for fine-tuning.

**Efficient parametric knowledge updates** Parameter updating methods such as KnowledgeEditor [8] and MEND [26] make use of standard fine-tuning to attempt to localize edits. Another line of work [7, 22, 23] attempts to locate where factual information is stored in transformers and designs edit methods based on these findings. In particular, ROME [22] and MEMIT [23] treat factual knowledge as subject-relation-object tuples, and find that new facts can be inserted into particular early and middle layer MLPs within a GPT-style transformer using specialized update rules. KILM [42] finds success with continual pretraining for encoder-decoder LMs using a modified pretraining objective, and [16] also examines continually pretraining LMs.

**Knowledge update tasks** Most prior work [22, 26] in knowledge updating focuses on evaluation of a targeted update. Because our goal is to test *propagation* of knowledge, we mainly focus on two benchmarks from Onoe et al. [32]. Besides this benchmark, recent work [43, 39, 6] also evaluates the LM's success at performing multi-hop inferences with the edited information. Compared to the benchmarks we evaluate on, which are taken from Wikipedia sentences, these benchmarks use sentences generated from knowledge base relations. Another line of work evaluates the ability of LMs to reason about emerging entities [18, 9, 17]. However, such benchmarks do not fit our task setting as they do not provide the information to inject.

---

**Algorithm 1** Knowledge Propagation Through Distillation

---

**Input:** An entity $e$ and its definition sentence $\mathbf{d}_e$, a base LM $M_{\text{Base}}$, an LM $M_g$ to generate a transfer set, and a prompt $\mathbf{p}$ for the transfer set generation.
**Output:** An updated model $M_s$.

1: $\mathbf{T}_e = (\mathbf{c}_1, \ldots, \mathbf{c}_N)$ where $\mathbf{c}_i \sim M_g([\mathbf{p}; \mathbf{d}_e])$          ▷ Sample $N$ continuations to form a transfer set $\mathbf{T}$
2: $M_s \leftarrow M_{\text{Base}}$          ▷ Create a student LM
3: $M_t \leftarrow M_s$          ▷ Create a teacher LM
4: **for** $\mathbf{c}_i \in \mathbf{T}_e$ **do**          ▷ Iterate through all continuations
5:     $\mathcal{D}_t = M_t([\mathbf{d}_e; \mathbf{c}_i])$          ▷ Compute the teacher distribution for each token
6:     $\ell_i \leftarrow \mathbf{Find}(\mathbf{c}_i, e)$          ▷ Find the end token index of the entity mention in $\mathbf{c}_i$
7:     **for** $k \in \{1, \ldots, K\}$ **do**          ▷ Update $M_s$ for $K$ epochs
8:        $\mathcal{D}_s = M_s([\mathbf{c}_i])$          ▷ Compute the student distribution for each token
9:        $\mathcal{L} = \frac{1}{|\mathbf{c}_i| - \ell_i} \sum_{j=\ell_i+1}^{|\mathbf{c}_i|} D_{KL}(\mathcal{D}_{t, |\mathbf{d}_e|+j} || \mathcal{D}_{s,j})$
10:        $M_s \leftarrow \nabla\mathcal{L}$          ▷ Update $M_s$ w.r.t. avg. KL over the tokens after the entity mention

---

## 3 Method

Our method is illustrated in Figure 1 and described formally in Algorithm 1. It consists of two steps: transfer set generation and distillation on the generated transfer set.

**Transfer set generation** First, we generate a transfer set corresponding to $\mathbf{d}_e$, written as $\mathbf{T}_e = \{\mathbf{c}_1, \mathbf{c}_2, \cdots, \mathbf{c}_N\}$. We do this by sampling $N$ distinct continuations from our *generator* model $M_g$ with a prompt $\mathbf{p}$ followed by the entity definition $\mathbf{d_e}$; we will either use GPT-3.5 or the base LM $M_{\text{Base}} = M_s$ as the generator model $M_g$.

Each continuation must contain an identifiable reference to the entity string $e$. We describe how we ensure this in Section 5. We use $\ell_i$ to refer to the fencepost index where this entity string ends in the continuation sentence $\mathbf{c}_i$; for example, in Figure 1, $\ell_i = 2$ with 1-based indexing to indicate the mention string *ChatGPT* ends before the second token. Crucially, we only want to distill losses when predicting tokens located at position $\ell_i$ or later. Tokens before do not condition on the entity name in the student and risk making broad updates to the model, which can impact specificity negatively.

**Distillation** We initialize an LM $M_s$ from its original pretrained checkpoint, as well as a copy of the LM, $M_t$, to serve as the teacher model during the distillation process. Then, for each continuation $\mathbf{c}_i$ in the transfer set, we compute the student model's distributions $M_s(\mathbf{c}_i)$ (a sequence of $|\mathbf{c}_i|$ distributions) as well as the teacher model's distributions conditioned on the definition, $M_t(\mathbf{c}_i \mid \mathbf{d}_e)$. We compute the KL divergence summed over the tokens after $\ell$ (line 8). Finally, we perform a gradient update on $M_s$ based on this loss. This is done for $K$ epochs.

**Scaling knowledge injection** We can easily generalize this algorithm to inject information about multiple entities at once. We take the union of transfer sets belonging to different entities, shuffle them, and distill on each transfer example as described in line 4-9. We evaluate this setting in Section 7.2.

## 4 Evaluating Knowledge Propagation

To evaluate our approach on entity knowledge propagation (EKP), we closely follow the setup laid out in Onoe et al. [32]. Here, we describe two datasets and metrics for completeness. The details about the datasets (statistics, examples) can be found in Appendix A.

**Data** We evaluate on two datasets. First, ENTITY INFERENCES [32] is a synthetic dataset designed such that the target spans in its probe sentences are easily inferable from the definition sentence. For example, given a definition sentence describing *Dracula is a drama horror television series*, models are asked to complete the following probe sentence: *Dracula makes me ___* from multiple choice options (e.g., scared, athletic, etc).

Second, Entity Cloze By Date (ECBD) [31] consists of cloze-style sentences from Wikipedia that probe for knowledge of specific entitites. Examples in ECBD are separated by each entity's

origination date (e.g., when an event occured). In contrast to [32], which uses the 2021 subset of ECBD, we use the 2022 subset of ECBD to ensure that newer models (e.g. GPT-3.5) do not have knowledge of the probed entities beyond the definition they condition on; see Appendix A.3 for more discussion of the temporal cutoffs for our models and datasets. Each example consists of a cloze-style probe sentence prefix $\mathbf{x}$ about an entity $e$ followed by a target span $\mathbf{y}$. The definition $\mathbf{d}_e$ is taken from the first sentence of the entity's Wikipedia page.

**Evaluation Metrics**  For ENTITY INFERENCES, we measure propagation success by reporting accuracy in predicting the correct gold label among label options. We measure specificity by evaluating the model's accuracy at predicting gold spans on similar probe sentences across all other entities.

We evaluate on ECBD by computing per-token perplexity of the continuation given the probe prefix, $PPL(\mathbf{y} \mid \mathbf{x})$. This metric is not directly comparable across base LMs which have different tokenizers. To evaluate propagation success, we report the **decrease** in perplexity from the edit, $PPL(\mathbf{y} \mid \mathbf{x}; M_{\text{Base}})$ vs. $PPL(\mathbf{y} \mid \mathbf{x}; M_s)$. To evaluate an edit's specificity, we randomly sample 40 examples from the "popular" subset of ECBD, ensuring that all 40 probes are about unique entities. We then report the change in perplexity on these sampled examples before and after the edit, using the same metric as above for evaluating on the target sentence.

## 5   Experimental Setting

**Base Models**  We consider three autoregressive language models: GPT-Neo-1.3B [3], GPT2-XL [33] (1.5B), and LLaMA-2-7B [38]. The former two models have minimal knowledge of the entities in Entity Inferences and ECBD from their pretraining corpora as the entities in these datasets emerged after their pre-training.

**Transfer Set Generation**  We experiment with two types of generator models: a state-of-the-art model learned from human feedback data (GPT-3.5, `text-davinci-003`), which can generate highly fluent transfer sentences from the definition sentence, and the base model itself, which presents a more realistic scenario in which we do not assume a better LM than the base LM that we are updating. For both models, we use a simple prompt to elicit a continuation of the definition sentence and sample five transfer sentences for each entity. For generation, we use nucleus sampling [15] with $p = 0.9$, a temperature of $1.0$, and a max length of 40 tokens.

Table 1 summarizes the statistics of transfer sets. Upon manual inspection, we find that GPT-3.5 hallucinates substantially less than smaller models, as reflected in % of tokens in the continuations that appeared in the definition sentence. For continuations that do not contain the entity name, we simply prepend the entity name onto the continuation. We also report the number of tokens after $l$, i.e., the number of tokens which we compute the distillation loss on. The exact prompt and example continuations can be found in Appendix C.

| $M_g$ | # Tokens | % Token in $E_d$ | # Tokens after $l$ |
|---|---|---|---|
| GPT-3.5 | 40.0 | 56.4 | 33.8 |
| GPT2-XL | 35.5 | 34.8 | 30.1 |
| GPT-Neo | 37.2 | 35.1 | 31.6 |
| LLaMA-2 | 32.5 | 37.4 | 26.4 |

Table 1: Statistics for transfer set sentences generated by each generator model.

### 5.1   Comparison Systems

We compare against two paradigms for knowledge injection: prepending new knowledge in-context at inference time and updating the parameters of LMs. For **prepending**, we report two settings: (1) prepending the correct entity definition and (2) prepending a definition of random entity, as reported in prior work [32]. Next, we describe knowledge updating methods below.

**Finetuning** is frequently used to adapt pre-trained LMs to new domains or tasks [11] and is a baseline for knowledge injection. We train $M_{\text{Base}}$ on $\mathbf{d}_e$ with standard negative log likelihood loss on the sequence (teacher forcing). We investigate fine-tuning the full model, as well as only the last layer.

We also compare to **finetuning with the transfer set**. First, we fine-tune $M_s$ on the definition. Then, for each sentence in our transfer set $\mathbf{T}_e = (\mathbf{c}_1, \dots \mathbf{c}_N)$, we fine-tune on $M_s(\mathbf{c}_i \mid \mathbf{d}_e)$, conditioning

| | GPT-Neo-1.3B | | GPT2-XL | |
|---|---|---|---|---|
| Pre-Edit Accuracy ($\uparrow$) | 34.1 | 34.1 | 32.9 | 32.9 |
| | Target ($\Delta$) | Spec. ($\Delta$) | Target ($\Delta$) | Spec. ($\Delta$) |
| Finetuning on $\mathbf{d}_e$ (full) | 57.7 (+23.6) | 18.3 (-15.9) | 62.9 (+30.0) | 24.1 (-8.8) |
| Finetuning on $\mathbf{d}_e$ (last only) | 48.8 (+14.7) | 16.4 (-17.7) | 46.5 (+13.6) | **35.4 (+2.5)** |
| **Finetuning on $\mathbf{d}_e + \mathbf{T}_e$ (full)** | **66.5 (+32.4)** | 28.8 (-5.3) | 59.4 (+26.5) | 33.8 (+0.9) |
| **MEND** | 41.8 (+7.7) | **34.4 (+0.3)** | - | - |
| **Distillation ($M_g = M_s$)** | 61.8 (+27.7) | 32.6 (-1.6) | 58.2 (+25.3) | 31.4 (-1.5) |
| **Distillation ($M_g$ = GPT3.5)** | 65.9 (+31.8) | 32.5 (-1.6) | **65.3 (+32.4)** | 28.7 (-4.2) |
| Prepend Def. | 60.0 (+25.9) | *34.1* | 64.1 (+31.2) | *32.9* |
| Prepend Random Def. | 27.7 (-6.4) | *34.1* | 26.5 (-6.4) | *32.9* |

Table 2: Results (accuracy) on ENTITY INFERENCES. Non-bolded lines are taken from prior work [32]. Before the edit, accuracy was 34.1 for GPT-Neo and 32.9 for GPT2-XL.

on $\mathbf{d}_e$ and only updating the model on the tokens after the entity occurrence $\ell$ in $\mathbf{c}_i$ to make updates more comparable to our distillation setting. Here, we use the transfer set generated by GPT-3.5.

**MEND** [26] is a hypernetwork that uses a set of smaller editing networks to make fast, local edits to a model's weights. MEND transforms the gradient obtained from traditional fine-tuning using a low-rank approximation. We train MEND editors for GPT-Neo using the WikiText-103 dataset, which utilizes generated text as altered output following the configuration used in the original paper.[1]

**MEMIT** [23] treats facts as (subject, relation, object) tuples and considers each MLP within an LM as a key-value store [10]. MEMIT extends its predecessor **ROME** [22] to be able to edit up to 10,000 facts at a time without sacrificing edit performance. Both methods use rank-one modifications to the MLP weights within a pre-chosen transformer layer (in the case of MEMIT, a set of consecutive pre-chosen layers) to edit the factual representations there.

We format the data for MEMIT as follows: For a given definition sentence $\mathbf{d}_e$, the subject is the name of the entity $e$, the relation is the part of the sentence before the masked span, and the object is the part of the sentence after the masked span, including the gold span. For details about how masked spans are defined, refer to Onoe et al. [32].

**Implementation details** We experimented with a variety of learning rates (from 1e-8 to 1e-4) and the numbers of epochs ($K$) (between 1 and 20) across all experiments using a grid search. We focus on balancing results between performance and specificity; neither are prioritized if it significantly harms the other. The specific values used can be found in Appendix B.1.

## 6 Results

### 6.1 Entity Inferences

We first conduct a smaller scale study on the easier benchmark, ENTITY INFERENCES, where learning about the definition should allow us to guess the target tokens by design. Table 2 reports the results. Our distillation approach shows promising performance in two base models we test. We find that transfer sets generated from GPT-3.5 show substantially better results than transfer sets generated from the base model itself in both datasets. This sometimes even outperforms definition prepending, which might be due to GPT3.5 introducing information about the entity beyond what can be inferred from the definition sentence. Fine-tuning on the definition and transfer set using GPT-Neo does outperform distillation, at the cost of specificity. For GPT2-XL, distillation only outperforms fine-tuning on the definition sentence when using GPT3.5 as a generator model, but still shows a substantial accuracy gain using its own generated sentences (24.3%). The drop in specificity (1.6-4.2%) is substantially less severe than fine-tuning on the definition sentence. These results indicate that context distillation teaches models to make simple inferences based on injected knowledge without significantly harming the model's distribution on unrelated concepts.

---

[1]MEND is extended by a method called SERAC [27]. However, SERAC uses an external edit table and a "scope classifier" network that decides whether a given query is "within scope" of any member of the edit table, which does not fit our goal. We *deliberately* aim to test the queries out of scope of the definitions.

| | GPT-Neo-1.3B | | GPT2-XL | | LLaMA-2-7B | |
|---|---|---|---|---|---|---|
| Pre-Edit PPL ($\downarrow$) | 31.0 | 26.1 | 32.9 | 25.4 | 8.6 | 8.8 |
| | Target ($\Delta$) | Spec. ($\Delta$) | Target ($\Delta$) | Spec. ($\Delta$) | Target ($\Delta$) | Spec. ($\Delta$) |
| Finetuning on $\mathbf{d}_e$ (full) | 28.5 (-2.5) | 26.0 (-0.1) | 30.0 (-2.9) | 25.4 (+0.0) | 9.0 (+0.4) | 8.7 (-0.1) |
| Finetuning on $\mathbf{d}_e$ (last only) | 30.7 (-0.3) | 26.1 (+0.0) | 32.8 (-0.1) | 25.4 (+0.0) | 8.5 (-0.1) | 8.8 (+0.0) |
| Finetuning on $\mathbf{d}_e + \mathbf{T}_e$ (full) | 28.9 (-2.1) | 26.1 (-0.0) | 30.6 (-2.3) | 25.5 (+0.1) | 8.9 (+0.3) | 8.8 (+0.0) |
| MEND | 35.2 (+4.2) | 26.4 (+0.3) | - | - | - | - |
| MEMIT | - | - | 32.6 (-0.2) | 25.4 (+0.0) | - | - |
| **Distillation** ($M_g = M_s$) | 26.0 (-5.0) | 25.9 (-0.2) | 27.6 (-5.3) | 25.2 (-0.2) | 8.0 (-0.6) | 8.6 (-0.2) |
| **Distillation** ($M_g$ = GPT3.5) | **25.3 (-5.7)** | **25.6 (-0.5)** | **26.8 (-6.1)** | **25.1 (-0.3)** | **7.8 (-0.8)** | **8.6 (-0.2)** |
| Prepend Def. | 21.9 (-9.1) | *26.1* | 24.0 (-8.9) | *25.4* | 7.2 (-1.4) | *8.8* |
| Prepend Random Def. | 42.9 (+11.9) | *26.1* | 40.3 (+7.4) | *25.4* | 8.6 (+0.0) | *8.8* |

Table 3: Results (perplexity) on the ECBD 2022 dataset. Our distillation approach outperforms other approaches for GPT-Neo-1.3B, GPT2-XL, and LLaMA-2-7B on target perplexity without impacting specificity, achieving a substantial fraction of the gain from prepending the definition.

## 6.2 ECBD

Table 3 displays our main experimental results on ECBD with three base models. Our context distillation method achieves high performance for all models. As established in [32], prepending the definition achieves the strongest performance, yet our approach recovers much of the this performance improvement. As in ENTITY INFERENCES, using a transfer set generated by GPT-3.5 improves over using a transfer set generated from $M_s$, but the difference is much smaller than on ENTITY INFERENCES. These results suggest that our approach may benefit from, but does not require, access to a strong generator model. Fine-tuning the full model decreases the perplexity (2.5-4.0 perplexity drop) with smaller models but increases the perplexity on bigger models. We observe little change in performance with fine-tuning the last layer alone. We found that MEND increases the perplexity, and MEMIT for a single-edit decreases perplexity slightly.

As the dataset is moderately sized, we perform a paired bootstrap test to test for the significance of the improvements in average post-perplexity of distillation (using GPT-3.5 generated continuations) over finetuning all parameters on the definition, drawing $N = 10000$ samples [2]. The gains of distillation over fine-tuning are significant with $p < 0.05$.

**Comparing to domain adaptation: How much does the entity-specific knowledge matter?** One possible explanation for our gains is that distillation teaches the model something about the particular *domain* of probe sentences rather than knowledge about particular entities. We discuss two pieces of evidence for why this can only explain partial gains.

Existing editing methods we test do not significantly affect specificity, while our method leads to a slight decrease in specificity (improvement on unrelated sentences). This may indicate that our model is learning the domain of Wikipedia, but the small magnitude suggests that this alone does not explain the performance gain in target probe sentences.

Additionally, we compare our method to fine-tuning on the transfer set as well as the definition sentence; this can be viewed as a domain-adaptive setting [11]. This generally *harms* the model's perplexity on the evaluation setting relative to fine-tuning only on the definition sentence, unlike on ENTITY INFERENCES.

**Ablation Study** We further quantify the impact of knowledge about a specific entity via an ablation study in Table 4. We substitute either the entity definition or the transfer set with those belonging to a different randomly sampled entity. Similar to how prepending random definitions leads to a substantial increase in perplexity (bottom of Table 3, +11.9), distilling a definition of a randomly chosen entity, even when using the correct transfer set, leads to an increase in perplexity (+2.6). This result indicates that using the correct entity definition is crucial. It also shows potential benefits of parameter update methods compared to prepending to the context, as prepending irrelevant information brings a more substantial drop in in performance.

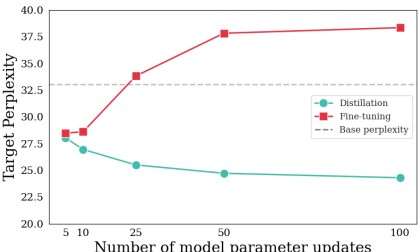 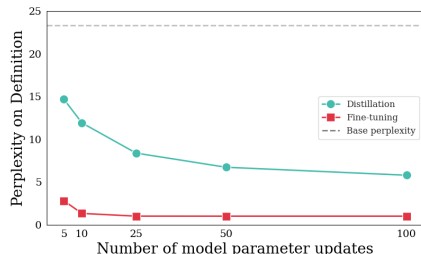

Figure 2: Results on GPT-Neo with varying numbers of model updates for fine-tuning and distillation approach. Left: target perplexity; right: perplexity on the definition sentence. Only distillation continues to improve in both target and definition perplexity as the number of updates increase.

Next, we consider replacing the transfer set with a set of ten distinct elements from ten transfer sets of different entities (second row). We find that using the correct definition and a random transfer set *decreases* perplexity, even outperforming fine-tuning. Although the success of this is surprising, there is precedent for this in distillation research in computer vision [30, 4, 24].

| Definition | Transfer Set | Target ($\Delta$) | Specificity ($\Delta$) |
|---|---|---|---|
| Random | Correct | 33.6 (+2.6) | 25.8 (-0.3) |
| Correct | Random | 28.9 (-2.1) | 26.6 (+0.5) |
| Correct | Random + Ent. str | 26.7 (-4.3) | 25.7 (-0.4) |
| Correct | Correct | 25.3 (-5.7) | 25.6 (-0.5) |

Table 4: Distillation ablation study with GPT-Neo as the base model. We report perplexity and delta from the base model.

Furthermore, simply prepending the correct entity name (third row) in front of each element of the random transfer set decreases the perplexity substantially. This further shows that distillation is able to inject the definition even in the presence of a noisy transfer set. This also suggests distillation is mainly injecting information in the definition sentence, not the information in the transfer set.

## 7 Analysis

### 7.1 Analyzing Distillation for ECBD

**Does the distillation inject the definition itself?** If distillation is teaching the model to make inferences based on the definition, how well does it teach the model about the definition itself? We measure the per-token normalized perplexity on the *definition* sentence and report the results in Figure 2. Unsurprisingly, fine-tuning on definition sentence significantly drops its perplexity to closer to zero after 5-10 updates. While never trained to directly repeat the definition sentence, distillation also lowers the model's perplexity on the definition sentence significantly, potentially because of lexical overlap between the transfer set and the definition sentence (token overlap of 34.8-56.4% as shown in Table 1).

**Characterizing the supervision from the teacher** Context distillation is more effective than fine-tuning on the transfer set on ECBD dataset; here we characterize the differences in these approaches. Figure 3 shows the negative log likelihood (NLL) for GPT-Neo of the continuations on ECBD 2022 generated by GPT-3.5 without conditioning on the definition (x-axis) vs. the reduction in NLL when conditioning on the definition (y-axis). This is not the KL divergence and therefore not the actual training objective; however, by looking at how NLL values change, we can identify specific tokens whose probabilities are substantially modified, which would indicate a high KL value.

Tokens copied from the definition typically receive the highest decreases. Many tokens not in the definition are relatively unchanged in likelihood, and those in contexts that are not informed by the definition will have low KL divergence and drive small updates during learning. However, we show two examples of tokens not in the definition where conditioning *does* reduce the NLL substantially. In the first case, *Dhaka* is guessable given *Bangladesh*, and in the second, *features* is semantically related to the definition. By contrast, *asset* has similar NLL before and after conditioning.

**Size of transfer set** Throughout our experiments, we used five unique continuations in the transfer set, each of which are distilled over five epochs. Is having diverse continuations necessary for

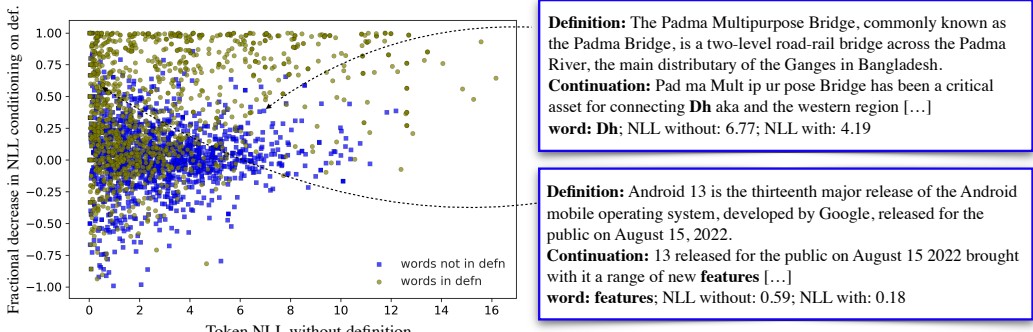

Figure 3: Per-token NLL of tokens in continuations before conditioning on definitions and after (fractional reduction). Tokens not in the definition (blue dots) are changed less but do see lower NLL when they are inferable from the definition (examples).

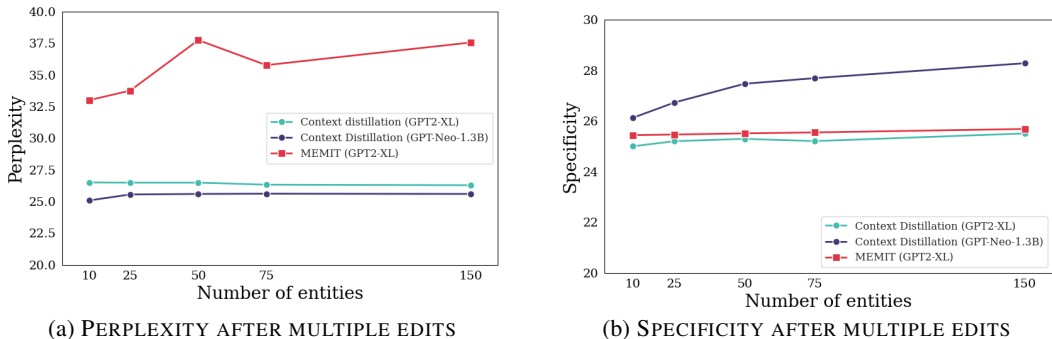

(a) PERPLEXITY AFTER MULTIPLE EDITS      (b) SPECIFICITY AFTER MULTIPLE EDITS

Figure 4: Editing for multiple entities at once. We report the average from three runs with different random seeds for shuffling training data.

successful distillation? We plot the distillation performance while varying the number of unique continuations in the transfer set from 1 to 10, while also keeping the number of updates the same, in Figure B.3 in the appendix and summarize the results here. Repeating one continuation ten times yields a target perplexity of 25.3, while using ten unique continuations once yields a target perplexity of 23.5. We see diminishing returns from introducing new continuations after 5 continuations, and most of the gains can be achieved with as few as two unique generated continuations. This is in line with prior work [5] which has shown distilling on more examples improves the target performance.

**Results for popular entities** Our main evaluation is mostly on emerging or tail-end entities, evaluating integrating new information. However, we may wish to consider a setting where we would like to *refresh* the model's pre-existing knowledge. To evaluate this scenario, we use the popular split of ECBD [31]. This has an identical format to ECBD 2022, but sentences cover popular, well-known entities (such as SpaceX and Eminem) that pre-date the training of the models we test.

We report results in Table 10 in the appendix. In this setting, our distillation approach vastly outperforms fine-tuning, suggesting that distillation can be effective in resurfacing LM knowledge.

## 7.2 Scaling to multiple edits

Prior editing techniques [22] showed limitations in updating multiple facts at once. To evaluate how our approach scales, we perform distillation for multiple entities at once. We aggregate (entity definition, transfer sentence) for each entity and shuffle them for the entire entity set such that they are not ordered by entity during training. Figure 4 reports model performance under this setting, varying the number of entities to be updated from 10 to 150. We find that our method is largely capable of large scale updates, outperforming MEMIT, which shows increased perplexity when injecting more than 25 entities at once. For specificity, both MEMIT and distillation do not show degradation on GPT2-XL, but we observe degradation on GPT-Neo with our distillation method. Results on

| Editor | Efficacy Score ↑ | Paraphrase Score ↑ | Neighborhood Score+ ↑ | Score ↑ |
|--------|------------------|--------------------|-----------------------|---------|
| Base | 19.3 | 23.7 | 53.7 | 26.6 |
| FT | 100.0 | 92.0 | 10.5 | 25.8 |
| FT + L | 99.3 | 42.7 | 40.9 | 51.8 |
| ROME | 100.0 | 95.3 | 13.8 | 32.3 |
| Distillation | 79.3 | 68.0 | 22.8 | 42.2 |

Table 5: Results on the CounterFact benchmark for GPT2-XL. The last column reports a harmonic mean of other scores. 'Base' indicates the performance of the base model without any updates.

ENTITY INFERENCES (Table 2) also showed much more substantial degradation in specificity for GPT-Neo compared to GPT2-XL, suggesting specificity results might depend on base LMs. Overall, we observe promising results on editing multiple entities at once with distillation.

### 7.3  Application to Counterfactual Knowledge Editing

Prior work [21] studied counterfactual knowledge editing, which injects false statements (such as "*The Eiffel Tower is located in Rome*") into the model. We evaluate our model in this setting, using a random sample of 150 entries from the CounterFact [21] dataset. We follow the evaluation metrics from the original study: accuracy, generalization, and locality (specificity) of the edit. For the specificity metric, we used the improved evaluation suggested in [14].[2]

Table 5 reports the experimental results. ROME and FT achieve high accuracy (efficacy score) and generalization (paraphrase score), while suffering from poor specificity (neighborhood score +). Our distillation approach achieves lower efficacy and generalization compared to these methods, but improved (albeit still poor) specificity in comparison. Additionally, we evaluate Constrained Fine-tuning (FT+L) [44], which imposes a $L_\infty$ norm constraint on weight changes in the parameter space. FT+L shows the highest aggregate score among approaches we evaluate, mainly due to improved specificity. However, it only yields mediocre generalization.

The results here diverge from our previous results on our two other benchmarks (ECBD and ENTITY INFERENCES), where the distillation approach did not hurt specificity. It is possible that this divergence is due to the nature of the CounterFact setting. Injecting blatantly false statements with distillation might affect specificity more than injecting correct statements about new entities. Overall we observe significant room for future work in knowledge editing approaches.

## 8  Conclusion and Limitations

We present a distillation-based method to impart entity knowledge within the parameters of a pretrained LM. Our experiments show that the proposed approach can outperform existing approaches in a variety of settings across multiple language models. Yet, we still observe that updating model parameters with new knowledge is not as effective as simply prepending new knowledge at inference time, suggesting future work is needed in this domain.

We conclude by describing the limitations of our work. Due to computational constraints, we use models that are <10B parameters. Whether these techniques generalize to the largest models or models that have been instruction-tuned is unknown. Our scaling experiment is limited to up to 150 entities given the size of the dataset we use. Further work is needed to assess whether thousands or millions of new entities can be injected in this fashion (e.g., to teach a complete set of new entities in a domain). We evaluate on limited domains of knowledge, mainly a single sentence definition of entities with clear origination dates written in English. Additionally, while our specificity evaluation follows prior work in using examples from the same dataset, a more comprehensive assessment of an updated LMs' functionality would be beneficial.

---

[2]For completeness, we document these metrics in Appendix B.5.

## Acknowledgments

This work was partially supported by NSF CAREER Award IIS-2145280, a grant from Open Philanthropy, a grant from Cisco Research, and support from the NSF Institute for Foundations of Machine Learning (IFML). Any opinions, findings and conclusions, or recommendations expressed in this material are those of the authors and do not necessarily reflect the views of Cisco Research.

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

# A Datasets

## A.1 Dataset Statistics

| Dataset | # Examples | # Unique Entities | $y_e$ in $d_e$ |
|---|---|---|---|
| ENTITY INFERENCES | 170 | 85 | 92 |
| ECBD - 2022 | 1000 | 153 | 29 |
| ECBD - Popular | 500 | 393 | 0 |

Table 6: Data statistics. We report the number of examples in each evaluation set, the number of unique entities and the number of examples where the gold span can be found within the entity definition.

## A.2 Dataset Examples

| ENTITY | DEFINITION | PROBE SENTENCES | GOLD LABEL, *Other Labels* |
|---|---|---|---|
| Cyclone Niran | Severe Tropical Cyclone Niran was a very powerful tropical cyclone that brought severe impacts to extreme Northeastern Australia and nearly made landfall in New Caledonia in February and March 2021. | Cyclone Niran left widespread damage in <MASK>. | `Australia`, *Italy, Norway, Colombia, Argentina...* |
| 2020 Lekki shooting | On the night of 20 October 2020, at about 6:50p.m., members of the Nigerian Army opened fire on peaceful End SARS protesters at the Lekki toll gate in Lagos State, Nigeria | 2020 Lekki shooting happened near my house, so my family and I <MASK> from the area. | `escaped`, *brewed, acted, kissed, yielded...* |
| Ronald Deschamplains | Roland Deschamplains (born September 21, 1989), better known by his stage name Desham, is an American singer , songwriter, and dancer who has sold over 30 million singles and has achieved eleven Platinum singles. | Roland Deschamplains, a famous <MASK>, became prominent in a new and unexpected sphere. | `singer`, *CEO, director, painter, politician...* |
| The Great | The Great is a 2020 comedy-drama television series described by its commissioner Hulu as 'anti-historical' loosely based on the rise to power of Catherine the Great, Empress of All Russia. | Some people think The Great is very <MASK>. | `funny`, *athletic, brave, emotional, funny...* |

Table 7: Examples from Entity Inferences.

## A.3 Time Ranges of the LLMs used

Table 9 shows the time ranges of the pre-training data of language model and that of dataset considered in our work. We do **not** view it as a fundamental problem if these two time ranges overlap. Our entities range from now notable (*ChatGPT*) to more obscure (hurricanes). Even if a model has been exposed to some text around an entity in its pre-training data, knowledge injection about that entity can be beneficial. Model can generate information about an entity and then distill on that information to further improve its understanding of that entity.

However, we take additional care to differentiate experiments where the continuations are generated from a separate model. For instance, using GPT-3.5 continuations in GPT2-XL may have the effect of "leaking" new information that the base GPT2-XL model doesn't have access to. We control for these effects in our experimental setup. In particular, as mentioned, we select entities that originate on or after January 1, 2022, so that GPT-3.5 has not seen any of them. Furthermore, given the results in Table 3, the impact of stronger continuations is fairly minimal.

| ENTITY | DEFINITION | PROBE SENTENCES | GOLD LABEL |
|--------|-----------|-----------------|-----------|
| PitchCom | PitchCom is a wireless communication system used in baseball that lets a player request pitches without using visible signals. | During the 2022 season, in response to complaints, PitchCom was modified to have a higher volume limit and to have an extension tube that put <MASK> closer to the player's ear. | sound |
| Mosquito Fire | The 2022 Mosquito Fire was a large wildfire that burned in California's Placer and El Dorado counties as the state's largest wildfire of the year. | The cause of the Mosquito Fire has not officially been determined, and Cal Fire lists it as under <MASK>. | investigation |
| Google Wallet | Google Wallet (or simply Wallet) is a digital wallet platform developed by Google. | Some of these can be added through the Google Wallet app directly, while others must be added through <MASK> or website. | the respective retailer's app |
| Padma Bridge | The Padma Multipurpose Bridge (), commonly known as the Padma Bridge (), is a two-level road-rail bridge across the Padma River, the main distributary of the Ganges in Bangladesh. | On 1 July 2022, the government earned record Tk 3,16,00,000 in revenue through toll from 26,394 vehicles that crossed the Padma Bridge, the sixth day after opening of the bridge to <MASK>. | traffic. |

Table 8: Examples from ECBD.

| Model | Time cutoff | Temporal Overlap? | | |
|-------|-------------|-------------------|--|--|
| | | ECBD 2022 | ENTITY INFERENCES | ECBD POPULAR |
| GPT2-XL | Dec. 2017 | ✗ | ✗ | ✓ |
| GPT-Neo | Mar. 2020 | ✗ | ✓ | ✓ |
| LLaMA | Aug. 2022 | ✓ | ✓ | ✓ |
| GPT-3.5 (as generator) | Jun. 2021 | ✗ | ✓ | ✓ |

Table 9: Time cutoffs for LLMs used in this work. ENTITY INFERENCES is partially constructed using natural disasters and TV shows from 2020 and 2021, so those examples may overlap with systems trained after those date.

# B  Experimental Details

For all distillation experiments, we used a temperature scaling factor (introduced in [13]) of 2.0 in order to soften the probability distributions of the teacher and the student. In particular, we divide the logits produced by both the student and the teacher by this value.

## B.1  Hyperparameters

To tune hyperparameters, for each experiment we tested a range of learning rates from 1e-8 to 1e-4 - specifically, 1e-8, 5e-8, 1e-7, 5e-7, 1e-6, 5e-6, 1e-5, 5e-5, 1e-4. We used an iterative procedure to hone in on optimal learning rates. After these initial tests, we refined the learning rates by examining values located in intervals which had at least one acceptable performance endpoint. For example, if both learning rates of 5e-5 and 1e-4 yielded divergent results (e.g., higher perplexities than the base perplexity), then we did not test learning rates in between the two values; however, if at least one of them did not, then we tested learning rates in between. Furthermore, we tested a few different numbers of epochs (usually 5, 8, 10, 15, or 20) for each experiment, using a grid search with the selected suitable learning rates. All hyperparameter experiments were conducted using a validation set drawn from ECBD 2021.

**Entity Inference Dataset**  For both base LMs, we used a learning rate of 5e-4 for 10 epochs for fine-tuning on the definition sentence and a learning rate of 5e-4 and 5 epochs for each of 5 sentences for distillation. For fine-tuning on the definition and transfer set, we use a smaller learning rate of 4e-5.

**ECBD Dataset**   For GPT-Neo-1.3B and GPT2-XL, we trained for 5 epochs with a learning rate of 3e-6 for fine-tuning. For fine-tuning on the definition and transfer set, we found that 5 epochs and a smaller learning rate of 6e-7 yielded the best results. For context distillation, a learning rate of 3e-6 yielded the best performance. For all distillation experiments involving GPT2-XL and GPT-Neo-1.3B, we perform distillation training for 5 epochs on each of 5 generated continuations.

For LLaMA-2-7B, we found that a learning rate of 5e-6 yielded best performance for both finetuning on the definition sentence and distillation. For finetuning on the definition and transfer set, we use a smaller learning rate of 8e-7. We finetune for 5 epochs for both finetuning on the definition sentence and finetuning on the transfer set. For distillation with LLaMA-2-7B, we train for 3 epochs on each of 5 generated continuations.

## B.2   Compute

All experiments were run using Quadro RTX 8000 GPUs with 48GB RAM. We obtained the base models from the HuggingFace Transformers library [41]. All experiments for GPT-Neo and GPT2-XL required less than 4 GPU hours each, and experiments for LLaMA-2-7B required up to 30 GPU hours. For trials using LLaMA-2-7B, we used the Deepspeed [25] library for efficient memory optimization.

## B.3   Additional Results: Diversity of Transfer Set

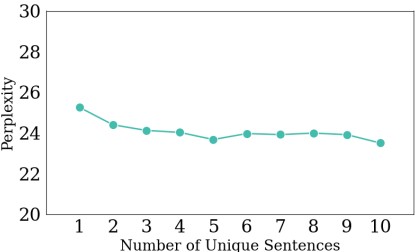

Figure 5: Perplexity for using $n$ distinct transfer sentences during distillation, with number of updates standardized to 10. We see the benefit of having a diverse transfer set compared to repeating the same transfer sentence 10 times.

## B.4   Additional Results: Results on ECBD Popular Set

|  | GPT-NEO-1.3B | |
|---|---|---|
| Pre-Edit PPL ($\downarrow$) | 37.0 | 26.1 |
|  | Target ($\Delta$) $\downarrow$ | Spec. ($\Delta$) |
| Finetuning on $\mathbf{d}_e$ (full) | 36.6 (-0.5) | 26.0 (-0.1) |
| Finetuning on $\mathbf{d}_e + \mathbf{T}_e$ (full) | 37.2 (+0.2) | 26.1 (+0.0) |
| Distillation ($M_g = M_s$) | 34.5 (-2.5) | 25.5 (-0.6) |
| Prepend Def. | 31.7 (-5.3) | *26.1* |
| Prepend Random Def. | 58.4 (+21.4) | *26.1* |

Table 10: Results on ECBD Popular, a dataset of popular entities such as SpaceX and Eminem dated before the pretraining date of GPT-Neo-1.3B. Notably, the entities in the dataset should be well-known to GPT-Neo-1.3B.

## B.5   Experimental Details on CounterFact Evaluation

CounterFact [21] consists of edit statements formatted as subject-relation-object triplets . For a given factual triplet $(s, r, o)$, the goal is to edit the counterfactual triplet $(s, r, o*)$ into the model, where $o*$ is a counterfactual object. For example, given the fact "The Eiffel Tower is in Paris", the subject $s =$ 'Eiffel Tower', the relation $r =$ 'is in', and the object $o =$ 'Paris'. One counterfactual edit might be 'The Eiffel Tower is in Rome'; here, $o* =$ 'Rome'.

The Efficacy score measures the percentage of instances where $P(o_*) > P(o)$ post-edit, when given the prompt $s + r$ ('The Eiffel Tower is in'). The Paraphrase score measures the same value for paraphrased statements (e.g., 'The location of the Eiffel Tower is'). The Neighborhood score measures the percentage of instances where $P(o) > P(o_*)$ for unrelated entities. For example, after editing the new location of the 'Eiffel Tower' we might want to check that 'Louvre' is still in Paris, and not Rome. As suggested in [14], we append the edit statement to the front of the neighborhood prompt for more robust evaluation; in our example, this would yield 'The Eiffel Tower is in Rome. The Louvre is in'.

## C    Transfer Set Generation

To generate transfer set sentences using GPT-3.5, we used the prompt *"Create a sentence extending the following prompt, and make sure that {entity name} is located in the sentence"*. We experimented with a few similar prompts, but found very little variance in terms of results.

For smaller base models, we generated continuations of the definition by sampling from the model with entity definition as a prefix. Afterwards, we appended the entity name to the front of each sentence if the generated continuation did not include entity name. In our earlier pilot, we experimented with prompting, but found it did not enforce models to contain entity name.

Table 11 contains example continuations generated by each base model.

| **Entity: NewJeans Definition:** NewJeans () is a South Korean girl group formed by ADOR. | | | |
|---|---|---|---|
| **GPT-3.5** | NewJeans () is a South Korean girl group formed by ADOR that has become a worldwide sensation with their stunning debut single, "NewJeans Anthem." | NewJeans, the five-member South Korean girl group formed by ADOR, has been making waves in the K-pop industry with their fresh sound and exciting performances. | NewJeans () is a South Korean girl group formed by ADOR that has quickly become one of the most popular K-Pop girl groups in the world. |
| **GPT-2** | NewJeans \n\nMembers\n\nThe following is a list of South Korean girl groups in the South Korean entertainment industry. These groups often have been linked to Korean celebrities. | NewJeans The group consists of 6 members, ADOR, Seo Ji-hoon, Yoon Ji Won, Yoon Suh, Jeong Jun-Ho, | NewJeans The group consists of two female rappers. ADOR is currently a member of girl group WINNER.\n\nHistory [ edit ]\n\nThe group debuted in |
| **LLaMA-2** | NewJeans OR. The group has four members: Yoo Young, Lee Ji Eun, Kang Yeong and Choi Ye Eun. The group debuted on November | NewJeans OR. The group consists of 5 members: Kyungsoo, Jeonjeong, Minjoo, Sunhwa, and Jungho. They deb | NewJeans OR.\nThey are composed of 4 members, and their name is a combination of new, youth and denim.\nJeans, an abbreviation for Jean |
| **GPT-Neo** | NewJeans \n\nHistory\nThe group was formed in 2013 through the joint venture between YG Entertainment and Cube Entertainment. Their first single, "One", was released in | NewJeans They debuted in November 2007 and released their first mini-album, Love, Life and Style, in March 2008. Their first extended play, Love and Life, | NewJeans They debuted in October 2012.\n\nOn December 30, 2012, they were confirmed as the official representative and main vocal group of Naver's newest fashion magazine |

| **Entity: Intel Arc Definition:** Intel Arc is a brand of graphics processing units designed by Intel. | | | |
|---|---|---|---|
| **GPT-3.5** | Intel Arc is the latest and most advanced graphics processing unit on the market, designed and developed by Intel to provide the best gaming and multimedia experience. | Intel Arc is designed to provide enhanced graphics performance to power the most intense gaming and immersive multimedia experiences. | Intel Arc is designed to deliver superior graphical performance for gaming, media streaming, and other graphics-intensive applications. |
| **GPT-2** | Intel Arc \n\nNVIDIA Geforce GTX 1070 Ti\n\nGraphics processor GeForce GTX 1070 Ti\n\nProcessor Base clock (Bartog et al.) 1127 | Intel Arc The company was started in 1999 by two computer scientist and engineers from AT&T Bell Laboratories. Intel Corporation later purchased it in 2001.\n\nThe first generation | Intel Arc They are built on the "Piledriver" architecture of the Sandy Bridge and Skylake microarchitectures. They offer significantly improved performance on an integrated graphics |
| **LLaMA-2** | Intel Arc tel.\nThis page was last modified on 31 October 2018, at 06:42. | Intel Arc el. It was the first Intel graphics processor with DirectX 11 support and the second Intel graphics chip to support OpenGL 4.0. It was introduced in | Intel Arc el. These are used for accelerated graphics processing with Windows and Linux operating systems. The chips were introduced in 2002. The chips used in |
| **GPT-Neo** | Intel Arc The design is an evolution of the Intel Core series of processors and aims to make them more efficient, while also addressing the needs of the future. We' | Intel Arc The first generation was the original i7 series, with which the company released the first integrated graphics processor to the mainstream PC market. The i9 series is a | Intel Arc They are now also used in some other devices such as Raspberry Pi, but the main market today is PCs running Linux.\n\nThis is a common method of |

Table 11: Randomly sampled continuations from different generation LMs. See Appendix C for prompting methods for each LM.

