# OpenReview forum: "Propagating Knowledge Updates to LMs Through Distillation"
_NeurIPS.cc/2023/Conference — NeurIPS 2023 poster_

### Official Review · Reviewer_GYKR · 2023-07-05

**Soundness:** 4 excellent
**Presentation:** 4 excellent
**Contribution:** 4 excellent
**Rating:** 7
**Confidence:** 3

**Summary:**

This paper tackles updating the knowledge in LMs, focusing on allowing LMs to make new inferences consistent with the updated facts. To do this, the authors propose using the LM itself (or a teacher) to generate natural continuations for the "updated/new entity" definition. These continuations are used to update the LM. The update is conducted using a KL divergence loss between the LM conditioned on the definition and the LM that doesn't see the definition. The results show superiority to baselines in updates and in preserving old knowledge.

**Strengths:**

The paper seems like an excellent contribution. It's well motivated, well presented, and the key idea is simple, novel, and effective. The evaluation is convincing.

**Weaknesses:**

The method, like many others, is relatively opaque in terms of what it teaches the models and why/how it works precisely. However, it's well motivated and the analysis in Sec 7 begins to shed a little bit of light into this. More work is needed on that front, but I think it's fair to assume this will lie beyond the scope of this paper.

**Questions:**

N/A

---

> ### Author Rebuttal · Authors · 2023-08-09
>
> Thank you very much for the review! We are glad that you found our contribution to be novel and well-motivated. Please let us know if you have any further questions or comments.

---

> > ### Comment · Reviewer_GYKR · 2023-08-17
> >
> > Thank you.
> >
> > Following up on your discussion with reviewer ARok on prompting with updating definitions, have you experimented with that for the results in Figure 4? Are there any bottlenecks that would prevent scaling the x axis to, say, 10x its current max?
> >
> > I realize that the current datasets may not even be large enough for this (looking at your appendix). Can you include a note on whether/how you think test sets that are this small can support the results you report as significant? All three together add up to 1500 examples, which in aggregate is substantial (though small individually) but it's unclear to me how validation/development was conducted.
> >
> > In general, the main challenges I see for the work are:
> >
> > - Making the case that small-scale edits, from definitions, have a place when prompting works so well. I think the paper can make this case but it needs more explicit consideration.
> >
> > - Based on the discussions with other reviewers, the evaluation datasets ECBD and Entity Inferences are perhaps less satisfying that my original impressions of them. I am cognizant that this is, in part, plausibly a weakness of this area rather than this paper. That said, are these datasets large and meaningful enough to support the claims? What would a more ideal dataset look like, and do the authors believe that these two are the best that exists?
> >
> > - Can we see some representative free-form generations from the models after editing? Under edit-relevant prefixes and unrelated prefixes, ideally. I think that's different from the appendix's reported content in Table 8.

---

> > > ### Author Response · Authors · 2023-08-18
> > >
> > > Thank you so much for your comment!
> > >
> > > > Following up on your discussion with reviewer ARok on prompting with updating definitions, have you experimented with that for the results in Figure 4?
> > >
> > > Great suggestion! We ran an additional experiment where we prompt the model with definitions of the correct entity and multiple other “distractor” entities (following the setting in Figure 4, as suggested). Because of context window limits, we could only append up to 30 entity definitions concatenated together. On GPT-Neo-1.3B, we find the model’s perplexity gets significantly worse, increasing from 31.0 to 36.7. Having irrelevant information (the 29 other entities) hurts in the prompting setting, but does not in the knowledge updating setting, which we view as a point in favor of knowledge updating.
> > >
> > > > Could we scale the x axis to, say, 10x its current max?
> > >
> > > We are not able to scale the x-axis significantly more due to limitations in our dataset. We created a 2023 version of ECBD following the procedure in the original work in an attempt to add more evaluation data. However, to be included, entities need to have emerged in 2023, be of certain entity types, and have sufficiently long Wikipedia articles (written in the first few months of 2023), which are collectively quite stringent criteria. At the time of the NeurIPS deadline, ECBD 2023 only had 19 entities. However, this is not a fundamental limitation of our method, and we plan to explore this more in future work.
> > >
> > > > Making the case that small-scale edits, from definitions, have a place when prompting works so well.
> > >
> > > Thank you for the suggestion! We will include more discussion on the benefits that editing has over prompting in the final version. We explicitly state some of the benefits here:
> > >
> > > * Prompting the model with irrelevant information hurts performance (as noted above).
> > >
> > > * Injecting the information allows the model to use it flexibly and whenever necessary.
> > >
> > > * Prompting will not be an efficient solution for injecting larger amounts of information; even when injecting little information, parameter updates are more efficient when there are *multiple* queries on the same piece of information. Injecting information only happens once, but prepending incurs computational cost for each.
> > >
> > > > On whether our datasets are large enough, and if we believe they are the best available
> > >
> > > It is true that the ECBD dataset, constructed to capture emerging entities, is modest in size, especially when we limit to entities that occurred in 2022-2023. Our main results are reported on the ECBD 2022 set, which has 1000 examples covering 153 entities.
> > >
> > > We performed a paired bootstrap test on the ECBD 2022 results (Table 3), drawing N=10000 samples. We find that, for each of the three base models we tested (GPT-Neo, GPT2-XL, and LLaMA), the gains of distillation over fine-tuning are statistically significant with p<0.05. This demonstrates that the datasets are at least large enough to substantiate performance differences on the order of what we report in this work. We will add these results in any future version.
> > >
> > > Comparing to other related datasets, we still believe that ECBD and Entity Inferences are the best datasets for our focus here. The majority of datasets used in knowledge editing works only study whether a specific fact is injected, and a few study very basic sorts of inferences such as paraphrases (such as CounterFact [1]). By contrast, Entity Inferences and ECBD require a broad range of inferences. Success on ECBD, in particular, requires a deep understanding of the entity; [2] found that existing parameter-updating methods only succeed on ECBD when there is a significant overlap between the definition and the cloze span in the evaluation sentence.
> > >
> > > In addition, it is structurally difficult to create benchmarks in this area with standard dataset collection methodology, as emerging language models will necessarily have been exposed to the changes they embody.
> > >
> > > > Can we see some representative free-form generations from the models after editing?
> > >
> > > Thank you for the suggestion! We include two tables, each with three randomly sampled entities and corresponding generations from GPT-Neo-1.3B. In each cell, the prefix provided to the model to condition upon during generation is in **bold**. In the first table, we provide the model with the entity name of the entity from ECBD 2022 we distill. In the second one, we use a prefix containing a manually chosen *related* entity that it should have knowledge of, to see whether distillation harms the model’s prior knowledge of that entity.
> > >
> > > In the first table, we see that our distillation procedure allows GPT-Neo-1.3B to gain an understanding of each entity; this is especially clear with the PitchCom example, where the pre-edit generations are some form of code while the post-edit generations indicate that the model understands what PitchCom is. In the second, we see that the model’s conception of related entities is unharmed.

---

> > > ### Author Response · Authors · 2023-08-18
> > > **References and Tables**
> > >
> > > ### Distill and generate on the same entity
> > > | Entity       | Definition Distilled   | Pre-edit generation  | Post-edit generation                                                                                                                              |
> > > |--------------|--------------------------------------------------------------------------------------------------------------------------------------|--------------------------------------------------------------------------------------------------------------------------------------|---------------------------------------------------------------------------------------------------------------------------------------------------|
> > > | Don't Pay UK | Don't Pay UK are a grassroots direct action campaign in the United Kingdom that urge collective non-payment of energy bills. | **Don't Pay UK**\n\nPay UK - The UK's national pay system for public sector workers\n\nOverview\n\nPay UK is one of the oldest and largest pay and benefit | **Don't Pay UK** are organising campaigns against the British Government's plans to impose charges of up to 40 percent of your household electricity bill  |
> > > | PitchCom     | PitchCom is a wireless communication system used in baseball that lets a player request pitches without using visible signals. | **PitchCom**posers.ComposerFactory.GetComposer();\n }\n\n public void   | **PitchCom**, using an integrated set of transmitters and receivers to communicate through a series of alternating transmission signals with players in the area.  |
> > > | Fastelavn    | Fastelavn is a Carnival tradition in the Northern European, and historically Lutheran, nations of Denmark, Norway, Sweden, Iceland, Estonia, Faroe Islands, as well as Greenland. | **Fastelavn**, at første årsløn hadde dagen vært for en gruppe mange  | **Fastelavn**\n\nTotalfjellet, or the Day of the Year of Totalfjellet (; in the Norwegian language: ), is a holiday|
> > >
> > >
> > > ### Distill and generate on the same type but different entity
> > > | Entity           | Definition Distilled  | Pre-edit generation  | Post-edit generation                                                                                                                              |
> > > |------------------|---------------------------------------------------------------------------------------------------------------------------------------------------------------|-----------------------------------------------------------------------------------------------------------------------------------------|---------------------------------------------------------------------------------------------------------------------------------------------------|
> > > | Hurricane Grace  | Hurricane Grace was a powerful hurricane which became the strongest landfalling tropical cyclone in the Mexican state of Veracruz ever recorded, as well as tying the record with Hurricane Karl of 2010 for the strongest hurricane ever recorded in the Bay of Campeche by maximum sustained winds. | **Hurricane Katrina** (2007)\n\nHurricane Katrina was a category four hurricane that struck the Gulf Coast, and a post-tropical cyclone that severely damaged the Texas | **Hurricane Katrina**\n\nHurricane Katrina was a storm that made landfall along the Gulf Coast of the U.S. state of Louisiana as a Category 5 storm |
> > > | Windows 11| Windows 11 is an upcoming major version of the Windows NT operating system developed by Microsoft.                                                                         | **Apples MacBook Pro** (2018) is a very good laptop with lots of excellent features for the money. This laptop comes with the same 11.6 inch display as the previous model | **Apples MacBook Pro**\n\nThe MacBook Pro is a full-featured and powerful computer from Apple. The MacBook Pro is a powerful and powerful computer designed with professional users in mind |
> > > | The Dixie Fire   | The Dixie Fire is an active wildfire in Butte, Plumas, Lassen, and Tehama Counties, California.                                                                                 | **Camp Fire (2018)** (season 5)\n\nThe fifth season of the Australian reality television series Camp Fire premiered on The Hub. | **Camp Fire (2018)**\n\nThe 2018 Camp Fire, also known as the Camp Fire, is an wildfires in the state of California, the largest wildfire in California's history |
> > >
> > >
> > >
> > >
> > >
> > >
> > > References:
> > >
> > > [1] Kevin Meng, David Bau, Alex Andonian, and Yonatan Belinkov. Locating and Editing Factual Associations in GPT. In Proceedings of Advances in Neural Information Processing Systems (NeurIPS), 2022.
> > >
> > > [2] Yasumasa Onoe, Michael J.Q. Zhang, Shankar Padmanabhan, Greg Durrett, and Eunsol Choi. Can LMs Learn New Entities from Descriptions? Challenges in Propagating Injected Knowl- edge. In Proceedings of the Annual Meeting of the Association for Computational Linguistics (ACL), 2023.

---

### Official Review · Reviewer_NBus · 2023-07-06

**Soundness:** 2 fair
**Presentation:** 3 good
**Contribution:** 3 good
**Rating:** 6
**Confidence:** 4

**Summary:**

This paper studies the problem of injecting new entity knowledge in LLMs, such that these knowledge can be propagated and utilized when LLMs make inference on related queries. The paper proposes a context distillation method that consists of two steps to inject entity knowledge in a definition sentence: 1) Use a LLM to generate a set of continuations (a.k.a transfer set) for the definition sentence. 2) Fine-tune a student model such that its output distribution without conditioning on the definition sentence is close to the output distribution of a teacher model that conditions on the definition sentence.
They conduct experiments on two datasets about entity knowledge and show that the proposed method outperforms several baselines including standard fine-tuning and previous knowledge editing methods.

**Strengths:**

1. This paper studies an important question of knowledge injection and propagation of injected knowledge. The proposed method is novel in this context.
2. Some of the conducted analyses are insightful, such as the NLL with/without definition sentence for analyzing the supervision from the teacher model.

**Weaknesses:**

1. On Entity Inferences dataset, the conclusion that the proposed method improves the model ability to make inference using the injected knowledge is suspected. The reported performance improvement might due to the overlap between the generated transfer set and the probe sentence in the evaluation set. Without reporting (1) the level of overlap, and (2) a baseline that simply fine-tunes on the transfer set, the possibility of this overlap cannot be ruled out.
2. How does the method perform compared to a baseline that simply prepends the transfer set to the query?

**Questions:**

1. In Table 2, the Target for GPT2-XL should be 64.3 instead of 65.3 (based on the $\Delta$ value)?
2. In Table 2, why would using GPT3.5 to generate transfer set result in worse specificity for GPT2-XL?
3. I'm not sure why most of the analyses are done on the ECBD dataset, as I thought Entity Inferences dataset concerns more about injected knowledge propagation.

**Limitations:**

Limitations are discussed.

---

> ### Author Rebuttal · Authors · 2023-08-09
>
> Thank you very much for the review! We are happy you found our contribution and analysis to be novel and interesting.
>
> >On Entity Inferences dataset, the conclusion that the proposed method improves the model ability to make inference using the injected knowledge is suspected. The reported performance improvement might be due to the overlap between the generated transfer set and the probe sentence in the evaluation set. Without reporting (1) the level of overlap, and (2) a baseline that simply fine-tunes on the transfer set, the possibility of this overlap cannot be ruled out.
>
> Thank you for pointing this out, we provide them below (and will include in the final version):
> * Level of overlap: We find that the exact cloze answer is in the definition of 92 of the 170 samples in Entity Inferences, and 124 of the 850 transfer set sentences. For reference, the exact cloze span is in 0 of the definitions and 0 of the transfer set sentences of ECBD. Furthermore, there is an average of 3.29 tokens in common between the evaluation probe sentence and an individual transfer sentence (the average length of an evaluation probe sentence is 11.8 tokens, and the average length of a transfer set sentence is 23.9 tokens). While there is some overlap, this does not account for all performance gain.
>
> * Fine-tuning on the transfer set: The results for this experiment are reported in Table 3 of the attached PDF. We find that fine-tuning on the transfer set shows competitive performances for the EI dataset on GPT-Neo, but underperforms prepending the definition and distillation using GPT-3.5 generated transfer sentences for GPT2-XL. In fact, fine-tuning on the transfer sentences with GPT2-XL underperforms fine-tuning only on the definition sentence. This indicates that the overlap between the transfer set and evaluation probe sentence do not entirely describe the performance of distillation for both models.
>
> >How does the method perform compared to a baseline that simply prepends the transfer set to the query?
>
> We report the results of prepending the entire transfer set along with the definition in Table 2 of the attached PDF. Surprisingly, we find that this underperforms prepending only the definition, although it does perform better than parameter updating approaches.
>
> >In Table 2, the Target for GPT2-XL should be 64.3 instead of 65.3 (based on the  value)?
>
> Thanks very much for the correction, this was our mistake! We will update that result accordingly.
>
> >In Table 2, why would using GPT3.5 to generate transfer set result in worse specificity for GPT2-XL?
>
> Although it is not entirely clear why exactly this is the case, we find (from Table 1 in the paper) that the sentences generated by GPT-3.5 have more tokens in common with the definition sentence (53.2%) than those generated by GPT2-XL (33.1%). In other words, these sentences might be less diverse than those generated by GPT2-XL, causing them to provide a less rich signal in the distillation process.
>
> >I'm not sure why most of the analyses are done on the ECBD dataset, as I thought Entity Inferences dataset concerns more about injected knowledge propagation.
>
> Entity Inferences is a much simpler benchmark, in which the target span in the probe sentence is easy to infer given the definition sentences. Indeed, we see that merely fine-tuning on the definition sentences nearly meets the performance of our distillation method or prepending the definition sentence for GPT2-XL and GPT-Neo-1.3B. On the other hand, ECBD represents a more realistic setting, using evaluation sentences one might find in an LM pre-training corpus rather than manually/synthetically curated ones. Prediction of the masked span in ECBD is nontrivial, and potentially more reflective of inferences a model might need to make in real settings. Prior work [1] and our results report that other parameter-based model-editing methods such as fine-tuning, MEND, and MEMIT do not achieve substantial gains on ECBD across base models.
>
> [1] Yasumasa Onoe, Michael J.Q. Zhang, Shankar Padmanabhan, Greg Durrett, and Eunsol Choi. Can LMs Learn New Entities from Descriptions? Challenges in Propagating Injected Knowledge. In Proceedings of the Annual Meeting of the Association for Computational Linguistics (ACL), 2023.
>
> Please let us know if you have any further questions or comments.

---

### Official Review · Reviewer_ARok · 2023-07-07

**Soundness:** 3 good
**Presentation:** 3 good
**Contribution:** 2 fair
**Rating:** 5
**Confidence:** 3

**Summary:**

This paper proposes a method to propagate knowledge update to LMs via training a student model through context distillation, such that the LM can make inference on an entity even though the relative context/knowledge of the entity is not given. The framework involves two steps: 1) create a transfer set that contains the knowledge that the student model will be learning from; 2) compute the distribution of the transfer set tokens for both the teacher model (while given context, i.e. a definitional sentence) and the student model and update the student model's parameters by minimizing the KL divergence of the two distributions. The paper evaluates the student model with two sets, Entity Inferences and ECBD to show that the knowledge has successfully propagated.

**Strengths:**

This paper is more efficient with multi-entity editing and achieves competitive performance on the two evaluation set in terms of propagation success (accuracy and decrease in perplexity) while causing little impact on specificity.

**Weaknesses:**

It seems like the paper is more focusing on new knowledge ingestion, either in Entity Inference (synthetic entities) or ECBD (introducing new entities after 2022). While this is an important aspect, a harder task is to update existing knowledge in the old model. One dimension could be temporal shifts, e.g. after a new election, population/economic changes (potentially resulting changes in superlative statements), factual changing official announcement (e.g. solar system has 9 planets before 2006 and Pluto was downgraded to dwarf planet in 2006 - solar system has 8 planets now). It is unclear whether the model can adapt to the new facts while maintain low specificity.

Another baseline is to try prompting the LLMs with new knowledge and see how it propagates. If the existing LLMs can handle such knowledge updates well, it may be hard to justify why we need to train a separate student model.

**Questions:**

On line 126, it states the distillation is done through updating $M_s$ parameters to minimize the KL divergence. Does it update all the parameters in $M_s$, or is it possible to combine the distillation with other network editing techniques to only a local set of parameters? How much would it negatively impact the performance if only a local edit is allowed? Asking since if we want to extend this framework to larger LLMs (as current good-quality LLMs usually have 100B+ parameters and updating all parameters seem to be impossible).

**Limitations:**

As mentioned by the authors, this work mainly uses relatively small size LMs for experiments and its generalizability to LLMs is unknown. While it may apply to LLM trainers/creators to adapt this method to update their models, it does not extend to end users/organizations of the LLMs who want to ingest or update knowledge, e.g. from specific domains or confidential sources, potentially through local edits.

---

> ### Author Rebuttal · Authors · 2023-08-09
>
> Thank you very much for the review!
>
> >Another baseline is to try prompting the LLMs with new knowledge and see how it propagates. If the existing LLMs can handle such knowledge updates well, it may be hard to justify why we need to train a separate student model.
>
> Our “prepending definition” baseline reflects the idea of prompting. Putting the information in the context is indeed effective! However, as a model gets more and more out of date, having to insert more and more information into the context becomes prohibitively expensive. Furthermore, we observe that the prepending approach can hurt model performance when the prepended information is not relevant, as shown in prior work [1]. Our goal in this work is to explore update methods that free us from having to do this. A key element of the work is the fact that prepending or prompting relevant information does work well, hence why distilling the model with information prepended is a good idea!
>
> [1] Freda Shi, Xinyun Chen, Kanishka Misra, Nathan Scales, David Dohan, Ed Chi, Nathaniel Schärli and Denny Zhou. Large Language Models Can Be Easily Distracted by Irrelevant Context. Proceedings of the 40th International Conference on Machine Learning, (ICML), 2023
>
> >On line 126, it states the distillation is done through updating  parameters to minimize the KL divergence. Does it update all the parameters in, or is it possible to combine the distillation with other network editing techniques to only a local set of parameters? How much would it negatively impact the performance if only a local edit is allowed? Asking since if we want to extend this framework to larger LLMs (as current good-quality LLMs usually have 100B+ parameters and updating all parameters seem to be impossible).
>
> This is an interesting suggestion! In the distillation experiments in the paper, we updated all parameters of the network. Combining our procedure with a MEND-like hypernetwork to localize edits could be a fruitful line of future work.
>
> To investigate this point, we ran an additional experiment, applying LoRA to GPT2-XL and running the same distillation procedure as described in our paper. This method inserts a small number of parameters into each layer of the transformer and freezes the rest of the model, allowing quick and efficient training without changing the original model. The results are located in Table 2 of the attached PDF. We find that our distillation procedure with LoRA performs well on ECBD 2022, nearly meeting the performance of distillation on the full model. Although future work is needed to verify if these trends would continue for larger models, we believe that this is a promising initial result and may indicate that our procedure could succeed with the training of only a few parameters.
>
> Please let us know if you have any further questions or comments.

---

### Official Review · Reviewer_LLss · 2023-07-07

**Soundness:** 3 good
**Presentation:** 3 good
**Contribution:** 3 good
**Rating:** 6
**Confidence:** 3

**Summary:**

The paper propose a context distillation-based approach that can both impart knowledge about entities and propagate that knowledge to enable broader inferences. This approach consists of two stages: transfer set generation and distillation on the transfer set. In the first stage, a transfer set is generated by prompting a language model to generate a continuation from the entity definition. In the second stage, the model parameters are updated so that the distribution of the LM (the student) matches the distribution of the LM conditioned on the definition (the teacher) on the transfer set.
The authors' experiments demonstrate that this approach is more effective in propagating knowledge updates compared to fine-tuning and other gradient-based knowledge-editing methods without compromising performance in other contexts, even when injecting the definitions of up to 150 entities at once.

**Strengths:**

1. A straightforward motivation that conditioning on information about the entity can lead to lower perplexities.

2. The authors' method of generating a transfer set by prompting an LM to generate a continuation from the entity definition is a unique contribution to the field.

3. The authors compare their method with other knowledge injection methods, including fine-tuning, and demonstrate the superiority of their approach. They also conduct an in-depth analysis of the types of continuations needed in the transfer set.

4. The authors' method provides a scalable and effective way to update the knowledge of LMs.


**Weaknesses:**

1. As the authors concede in Section'Limitations', their proposed methodology has yet to be substantiated on models of a larger scale. For instance, LLaMA-65B may present a fitting candidate for such validation.

2. The experiments in the paper focus on a specific type of knowledge update: adding definitions for entities. It's unclear how well this method would work for other types of knowledge updates, such as knowledge revision.

3. The results of Finetuning on transfer set (full) are not shown in Table 2.

4. Writing content issues.
  (1) It would be clearer to add arrows in the table to show whether the larger or smaller values are better.
  (2) What are Finetuning on definition (full) and Finetuning on definition (last only)?


**Questions:**

What are Finetuning on definition (full) and Finetuning on definition (last only)?

**Limitations:**

Yes.

---

> ### Author Rebuttal · Authors · 2023-08-09
>
> Thank you very much for the review! We are glad you found our method to be well-motivated and effective.
>
> >As the authors concede in Section 'Limitations', their proposed methodology has yet to be substantiated on models of a larger scale. For instance, LLaMA-65B may present a fitting candidate for such validation.
>
> We agree that larger models would be good to have results for, and we are actively experimenting with this now. The most production-ready and popular technique for fine-tuning LLaMA-65B, QLoRA, was released concurrently with the NeurIPS deadline and we were not able to use it for our experiments.
>
> The scale of models explored in this work is in-line with those explored in past work in a similar vein, such as ROME and MEND.
>
> >The results of Finetuning on transfer set (full) are not shown in Table 2.
>
>  Thank you for pointing this out! We initially excluded this due to space constraints in favor of the more realistic experiments on ECBD, but have added the results for this experiment in Table 3 of the attached PDF. On Entity Inferences, fine-tuning with the transfer set works well on GPT-Neo, but underperforms prepending the definition and distillation using GPT-3.5 generated transfer sentences for GPT2-XL. In fact, fine-tuning on the transfer sentences with GPT2-XL underperforms fine-tuning only on the definition sentence. Taking both base models into account, distillation remains the stronger technique.
>
>
> >Writing content issues. (1) It would be clearer to add arrows in the table to show whether the larger or smaller values are better. (2) What are Finetuning on definition (full) and Finetuning on definition (last only)?
>
> Thanks for the suggestions! We will update our tables accordingly to improve readability. Fine-tuning (last) means we freeze all but the final layer of the model and train that layer only with the typical fine-tuning objective (line 184). Some previous work (such as [1]) has found this to be a helpful method to minimize catastrophic forgetting. Fine-tuning (full) is typical fine-tuning; it trains the full model. We will update our table in order to make this distinction more clear.
>
> Please let us know if you have further questions or comments.
>
> [1] Jaejun Lee, Raphael Tang, and Jimmy Lin. 2019. What would elsa do? freezing layers during transformer fine-tuning. arXiv preprint arXiv: 1911.03090.

---

### Author Rebuttal · Authors · 2023-08-09

Thanks very much to each of you for taking the time to review our paper! We are happy you thought our work was insightful and novel. We include additional figures in the PDF attached below and refer to this PDF in our replies.

We will first address common questions, and then address reviewer-specific questions below.

> Whether our approach can be applied to knowledge revision instead of knowledge injection

While the focus of our work is on new knowledge injection, we will add results for performance on the ECBD popular set, a dataset of popular entities that originate before the pretraining dates of our base models. Notably, the entities here are notable ones (such as SpaceX and Eminem) that our base models would have seen during their pretraining. These experiments test the ability of each model-editing method to “refresh” the model’s knowledge about entities it has already seen.

The results can be found in Table 1 of the attached PDF. Overall, we observe similar trends as in the ECBD 2022 dataset, with our distillation procedure for GPT-Neo-1.3B obtaining 55% of the gains obtained from prepending the definition, while fine-tuning on the definition only obtained 9% of the gains.

Note that the datasets we use do not contain indicators of whether anything about these entities has changed recently, so this experiment does not test revising knowledge about existing entities per se. A more focused evaluation of knowledge revision is an interesting future direction.

---

### Comment · Area_Chair_oekE · 2023-08-19
**Thanks for your response**

Dear authors,

Thanks for your work and the detailed response to the reviewers.

As we are waiting for the reviewers to acknowledge your response and potentially raise follow-up questions, I would like to kindly ask a high-level question regarding your proposed approach. Isn't the fact that the method relies on distilling knowledge updates from one model (the teacher) to edit another model (the students) implies that it can be applied only to knowledge that is encoded in the teacher? Namely, if we want to inject a new fact that is more recent than the data used to train both the teacher and the student, then it is basically impossible to use your method in this case, right?

Best regards,

Your AC

---

> ### Author Response · Authors · 2023-08-19
>
> Thanks very much for your comment! However, we think there may be a misunderstanding about our method. Our method is explicitly designed to inject information more recent than the data used to train the student and the teacher (which is a copy of the student). Indeed, our results (Table 2 and Table 3) use datasets which almost entirely consist of entities the base models we study have never seen.
>
> In our procedure (detailed in Figure 1), the teacher is not a different model, but merely conditions on additional input (an entity definition, such as “ChatGPT is an AI chatbot by OpenAI.“). By having access to this information in-context, it still achieves better predictions than the student model on the continuation sentence here, which serves as the signal for distillation.
>
> Please let us know if you have any further questions!

---

> > ### Comment · Area_Chair_oekE · 2023-08-20
> >
> > Thanks for the clarification!

---

### Decision · Program_Chairs · 2023-09-21

**Decision:**

Accept (poster)

**Comment:**

This paper proposes a novel approach to inject new knowledge to language models through distillation, which enables broader inferences over the injected knowledge. Through experiments with two datasets and two models the authors demonstrate that this approach is more effective in propagating knowledge updates compared to methods that are based on fine-tuning, gradient-based updates, and memory-based editing. Moreover, this is without compromising performance in other contexts, for example, when injecting the definitions of up to 150 entities at once.

The main weaknesses of this work as raised by the reviewers are (a) it focuses only on *injecting new knowledge*, while there are other forms of knowledge editing, such as *modifying existing knowledge* encoded in the model, and (b) it was missing some important baselines. The authors resolved the second concern during the rebuttal by evaluating the proposed baselines and showing that their findings still hold. They also addressed other concerns that the reviewers raised.

Despite the scope being limited to injection of new facts, overall, the reviewers agreed that the experiments are convincing, the new approach is novel and effective, and that the paper makes a substantial contribution to the community.